# Stoichiometry-engineered phase transition in a two-dimensional binary compound

Mengting Huang[1,9], Ze Hua [2,9], Roger Guzman [3,9], Zhihui Ren[4], Pingfan Gu[5,6], Shiqi Yang [5], Hui Chen[1], Decheng Zhang[1], Yiming Ding[1], Yu Ye [5], Caizhen Li[4,7], Yuan Huang [1,8]✉, Ruiwen Shao [2]✉, Wu Zhou [3]✉, Xiaolong Xu [1]✉ & Yeliang Wang [1]✉

Due to complex thermodynamic and kinetic mechanism, phase engineering in nanomaterials is often limited by restricted phases and small-scale synthesis, hindering material diversity and scalability. Here, we demonstrate the exploration to unlock the stoichiometry as a degree of freedom for phase engineering in the Pd-Te binary compound. By reducing diffusion rates, we effectively engineer the stoichiometry of the reactants. We visualize the kinetic process, showing the stoichiometry transition from $Pd_{10}Te_3$ to $PdTe_2$ through a sequential multi-step nucleation process. In total, five distinct phases are identified, demonstrating the potential to enhance phase diversity by fine-tuning stoichiometry. By controlling spatially uniform nucleation and halting the phase transition at precise points, we achieve stoichiometry-controllable wafer-scale growth. Notably, four of these phases exhibit superconducting properties. Our findings offer insights into the mechanism of phase transition through stoichiometry engineering, enabling the expansion of the phase library in nanomaterials and advancing scalable applications.

Phase engineering is of paramount importance in numerous scientific and technological fields due to its ability to fundamentally alter the properties of materials[1–7]. The realization of phase engineering is more feasibly accomplished in nanomaterials, as it is capable of adjusting various parameters, such as temperature, pressure, and composition, in a highly controllable way through meticulous adjustment of the experimental conditions[1,2,8–10]. When the formation of the final structure of a nanomaterial is driven by the kinetic control of its synthetic process, a thermodynamically less favorable phase can be achieved. For example,

the metastable 1T' phase of $MoTe_2$ and $MoS_2$ are synthesized by engineering the kinetic process through precise control of the growth temperature and the choice of appropriate reaction precursors[11–16].

However, these phase transitions are typically confined within two phases, due to the lack of means to regulate the thermodynamic stability of other phases. Additionally, these phase transitions are restricted to situations where the chemical stoichiometry remains unchanged. The synthesis of solid-state compounds is often limited by the slow mass transfer rate and high activation energy in the

[1]School of Integrated Circuits and Electronics, MIIT Key Laboratory for Low-Dimensional Quantum Structure and Devices, Beijing Institute of Technology, Beijing 100081, China. [2]Beijing Advanced Innovation Center for Intelligent Robots and Systems, School of Medical Technology, Beijing Institute of Technology, Beijing 100081, China. [3]School of Physical Sciences, University of Chinese Academy of Sciences, Beijing 100049, China. [4]Centre for Quantum Physics Key Laboratory of Advanced Optoelectronic Quantum Architecture and Measurement, School of Physics, Beijing Institute of Technology, Beijing 100081, China. [5]State Key Laboratory for Mesoscopic Physics and Frontiers Science Center for Nano-optoelectronics, School of Physics, Peking University, Beijing 100871, China. [6]MIIT Key Laboratory of Semiconductor Microstructure and Quantum Sensing Department of Applied Physics, Nanjing University of Science and Technology, Nanjing 210094, China. [7]Beijing Key Lab of Nanophotonics and Ultrafine Optoelectronic Systems, Beijing Institute of Technology, Beijing 100081, China. [8]Advanced Research Institute of Multidisciplinary Science, Beijing Institute of Technology, Beijing 100081, China. [9]These authors contributed equally: Mengting Huang, Ze Hua, Roger Guzman. ✉e-mail: yhuang@bit.edu.cn; rwshao@bit.edu.cn; wuzhou@ucas.ac.cn; xuxiaolong@bit.edu.cn; yeliang.wang@bit.edu.cn

reaction[17–19]. Thus, the traditional use of high temperatures, long growth times and excessive reactants results in products with specific stoichiometry[20–23], which precludes the synthesis of less stable compounds with different chemical stoichiometries. This considerably affects the diversity and flexibility of phase engineering. However, the precise control of the chemical stoichiometry is challenging, as real systems are complex, encompassing both interdiffusion and nucleation, which are dependent on composition, defect concentrations, crystallographic orientations of the reactants[12,24–31]. Therefore, unlocking the freedom of stoichiometry, clarifying the kinetic nature of phase transition and thereby achieving the stoichiometry-controllable wafer-scale growth, which is a crucial bottleneck for the scaled-up applications of phase-engineered materials, is of great significance but remain challenging[2,5,32].

In this study, we showcase the exploration of unlocking the stoichiometry as a degree of freedom for phase engineering in the Pd-Te binary compound. We provide valuable perspectives into the mechanism of phase transition via stoichiometry engineering. Furthermore, by engineering the growth method to realize spatially uniform nucleation and to arrest the phase transition at exact points, we achieve stoichiometry-controllable wafer-scale growth. To slow down diffusion rates and capture the stoichiometry change process, we tellurize the Pd film at a temperature significantly below the melting points of all the materials involved. This approach allows us to observe a sequential phase transition with varying stoichiometries, achieved through a unique multi-step nucleation process. We further investigate the kinetic mechanisms, confirming that the phase transition is accompanied by a re-crystallization process, and highlight the crucial role of the hetero-phase interface during growth. The nucleation process acts as a barrier to regulate the final phase formation. To ensure uniform Te supply, a substrate coated with a Te film is placed face-to-face with the Pd substrate, enabling uniform nucleation and wafer-scale growth on various substrates. By controlling the cessation of the stoichiometry transition through the amount of Te involved in the reaction, stoichiometry-controllable growth is achieved. Four distinct stoichiometric phases are confirmed to possess superconducting properties. Our findings demonstrate the potential of utilizing stoichiometry as a degree of freedom to enhance the diversity of phase engineering in a binary compound, achieving stoichiometry-controllable wafer-scale synthesis and paving the way for scaled-up applications of phase-engineered materials.

## Results and discussion
### Multi-step nucleation
We facilitate the tellurization of the Pd film (-10 nm) deposited on the $SiO_2$ substrate by using an in-situ chemical vapor deposition (CVD) system. This system is equipped with a quartz observation window, allowing for real-time monitoring of the growth process via microscopy and capturing the kinetic phase transition process. To slow the diffusion rates and achieve stoichiometry modulation of the reactants, we reduce the growth temperature to 300 °C, which is significantly lower than the melting point of Te. The overall process can be classified into two steps: (1) the initial tellurization of the Pd film to form $Pd_{10}Te_3$, and (2) the stoichiometric phase transition in the Pd-Te system (Fig. 1a). The exposed cross section of Pd film at the edge of the substrate is more active and prone to adsorb Te atoms[33,34]. Nucleation begins at the substrate edges and gradually progresses inward (Fig. 1a). The contrast of the film undergoes multiple alterations, being accompanied by diverse nucleation, signifying the complex kinetic process (see Supplementary Movie 1 for details). By decelerating the diffusion rates, we disclose the sequential phase transition process, which is previously obscured by the high growth temperature and long growth time, resulting in only the $PdTe_2$ phase[20,35,36]. Compared with the direct one-step transition, multi-step phase transitions can effectively reduce the activation energy barrier from Pd to $PdTe_2$[26]. Totally,

we have identified five different stoichiometrics during the entire phase transition process.

To clarify the kinetic mechanism, we employ a rapid cooling approach to arrest the phase transition process. Figure 1b shows an optical image of the intermediate stage, showing several distinguishable regions. As nucleation initiates from the substrate edges, the optical images sequentially from left to right illustrate the temporal evolution of the nucleation and stoichiometric transition process. Figure 1c–e are enlarged images of characteristic regions, presenting the detailed features of different nucleations. Raman characterization of different nucleations reveals that the elliptical nucleations in Fig. 1d and the small circular nucleations in Fig. 1e (labeled by the purple and blue circles) respectively correspond to PdTe and $PdTe_2$ (Fig. 1h)[24,25,37]. The Raman scanning of PdTe nucleations presents uniform intensity, suggesting a complete phase transition (Fig. 1g). Other nucleations in Fig. 1c (labeled by red, orange and cyan circles) do not show significant Raman signals (Fig. 1f). Two of them are verified to be $Pd_{10}Te_3$, $Pd_9Te_4$ by the scanning transmission electron microscopy (STEM) characterization, which will be discussed in detail later. Between the $Pd_{10}Te_3$ and $Pd_9Te_4$ phase, there exists an intermediate phase (indicated by the orange dashed line in Fig. 1c), whose atomic ratio and lattice structure do not match any of the phases with known structures in the Pd-Te compound, signifying a previously unreported phase. Such stoichiometry-engineered phase transitions not only enhance the diversity of phase engineering but also offer an approach to discovering previously unreported phases.

As time progresses, the PdTe nucleations gradually enlarge and eventually merge to form a continuous film (see Supplementary Movie 1). Other phases undergo a similar process, though the shapes of the nucleation differ (Fig. 1b). When the thickness of the Pd thin film is reduced to 5 nm, only two kinds of contrasts changing with time are observed (see Supplementary Movie 2). Through Raman characterization, it is confirmed that these two contrasts are PdTe and $PdTe_2$ respectively (Supplementary Fig. 1). After the Pd film becomes thinner, the excessive Te in the environment causes the pioneering phases to be skipped directly or to be converted into the PdTe phase in an short time. The different phases possess excellent thermal stability. After annealing at 500 °C for 30 minutes, there is no obvious contrast change in the films (Supplementary Fig. 2). There is also no significant variation in the Raman signals of PdTe and $PdTe_2$ before and after annealing.

### Identification of stoichiometry of different phases
In order to identify the exact stoichiometry of the nucleations, we first utilize energy dispersive X-ray spectroscopy (EDX) to preliminarily identify the elemental composition and ratios in different nucleations (Supplementary Fig. 3). We take the identification of the PdTe and $Pd_9Te_4$ phases as an example for introduction. Figure 2a shows a low-magnification high-angle annular dark-field (HAADF) STEM image of PdTe nucleation. We carry out EDX scanning at the interface between the two phases. It shows a notable disparity in elemental content on both sides of the interface. Semi-quantitative EDX line scanning verified that the mole ratios of Te to Pd on the two sides are -1:1 and 7:3 respectively. In accordance with the Pd-Te binary phase diagram, these elemental ratios respectively correspond to PdTe and $Pd_9Te_4$ or $Pd_7Te_3$. We then obtain the atomic-resolution HAADF-STEM images (Fig. 2f–i) and selected area electron diffraction (SAED) patterns of different nucleation regions (Fig. 3b and Supplementary Fig. 4). Corresponding simulations are performed based on the hypothesized phases. By carefully analyzing the lattice parameters and symmetry of simulation results with the experimental results, we identify the phases as $Pd_{10}Te_3$, $Pd_9Te_4$, PdTe, and $PdTe_2$ (Fig. 2j–m), relevant information regarding the lattice structure shown in the Supplementary Table 1–4. The lack of Moiré fringes, which would typically emerge from the superposition of two distinct periodic structures, indicates that each of the grown phases is uniform in the vertical direction.

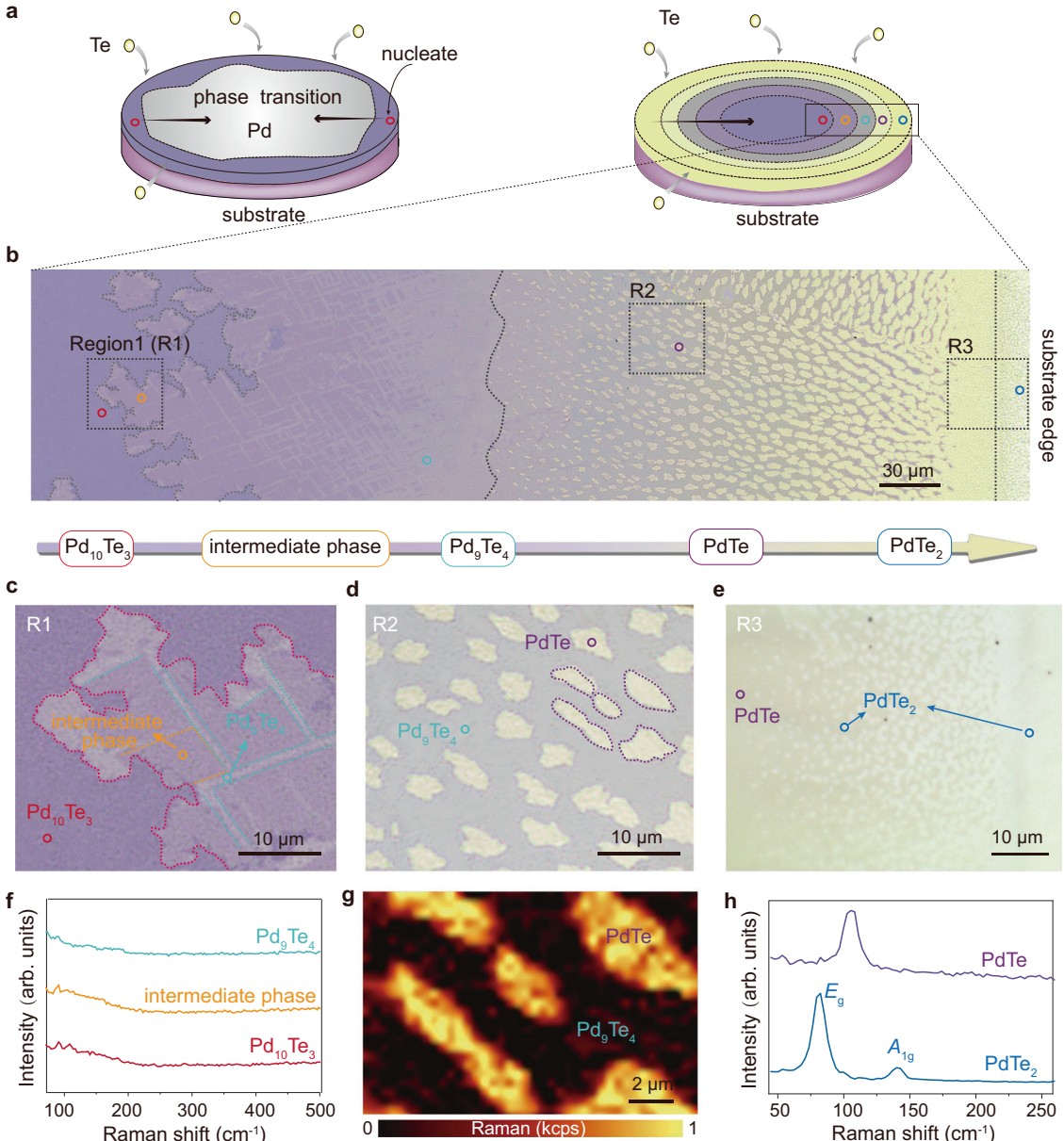

**Fig. 1 | Multi-step stoichiometric transition and diverse nucleation processes. a** Schematic diagram of the tellurization process of Pd films and their nucleation and phase transition. **b** Optical image of the intermediate stage of phase transition. The image from left to right corresponds to the center to the edge of the substrate, which can also be considered as the evolution of the phase transition over time. **c**–**e** Enlarged optical micrographs of nucleations of different phases. The intermediate phase, $Pd_9Te_4$, PdTe, and $PdTe_2$ phases exhibit nucleations with irregular, elongated, elliptical, and circular shapes, respectively. **f** The Raman spectra collected at different positions, marked by the points in Fig. 1c. No obvious Raman peaks are observed. This could be because the chemical bonds and molecular structures of these phases might result in relatively weak Raman scattering cross-sections. Source data are provided as a Source Data file. **g** The Raman spectral scan of PdTe nucleations exhibits a uniform Raman signal. **h** The Raman spectra characterization of the nucleation, indicated by the purple and blue dashed lines in Fig. 1d and e, respectively, reveals that the two types of nucleations correspond to PdTe and $PdTe_2$. Source data is provided as a Source Data file.

The nucleation with an irregular shape is an intermediate phase between $Pd_{10}Te_3$ and $Pd_9Te_4$ (Fig. 1c). As the stoichiometry transition proceeds towards a higher Te content, the Te content in the intermediate phase should be situated between that of $Pd_{10}Te_3$ and $Pd_9Te_4$, consistent with the EDX result. This phase shows a unique zigzag-like atomic structure (Supplementary Fig. 5). Currently, there is no lattice structure information in the database that matches the chemical stoichiometry and atomic arrangement of the intermediate phase. We conjecture that this phase could be a previously unknown phase.

The sequential stoichiometry transition between adjacent two phases is realized through nucleation in various shapes. The nucleation of the intermediate phase is in an irregular shape, while the $Pd_9Te_4$,

PdTe, and $PdTe_2$ phase nucleations show elongated strips, ellipses, and circles, respectively (Fig. 1c–e). According to the kinetic Wulff construction, the shape of the nucleation is determined by thermodynamic and kinetic factors, such as interface energy and the relative growth rate of different facets[38,39]. We then obtained atomic-resolved STEM images of the interface between two adjacent phases (Figs. 2c and Supplementary Figs. 6–8). The lattices of every two phases are seamlessly connected with atomically sharp boundaries.

At the interface between $Pd_9Te_4$ and PdTe, the short axis of the PdTe nucleation is parallel to the direction of the $Pd_9Te_4$ atomic chains (indicated by the red-colored arrow in the Fig. 2a, c), signifying that anisotropy of the lattice has an obvious influence on the rate of the

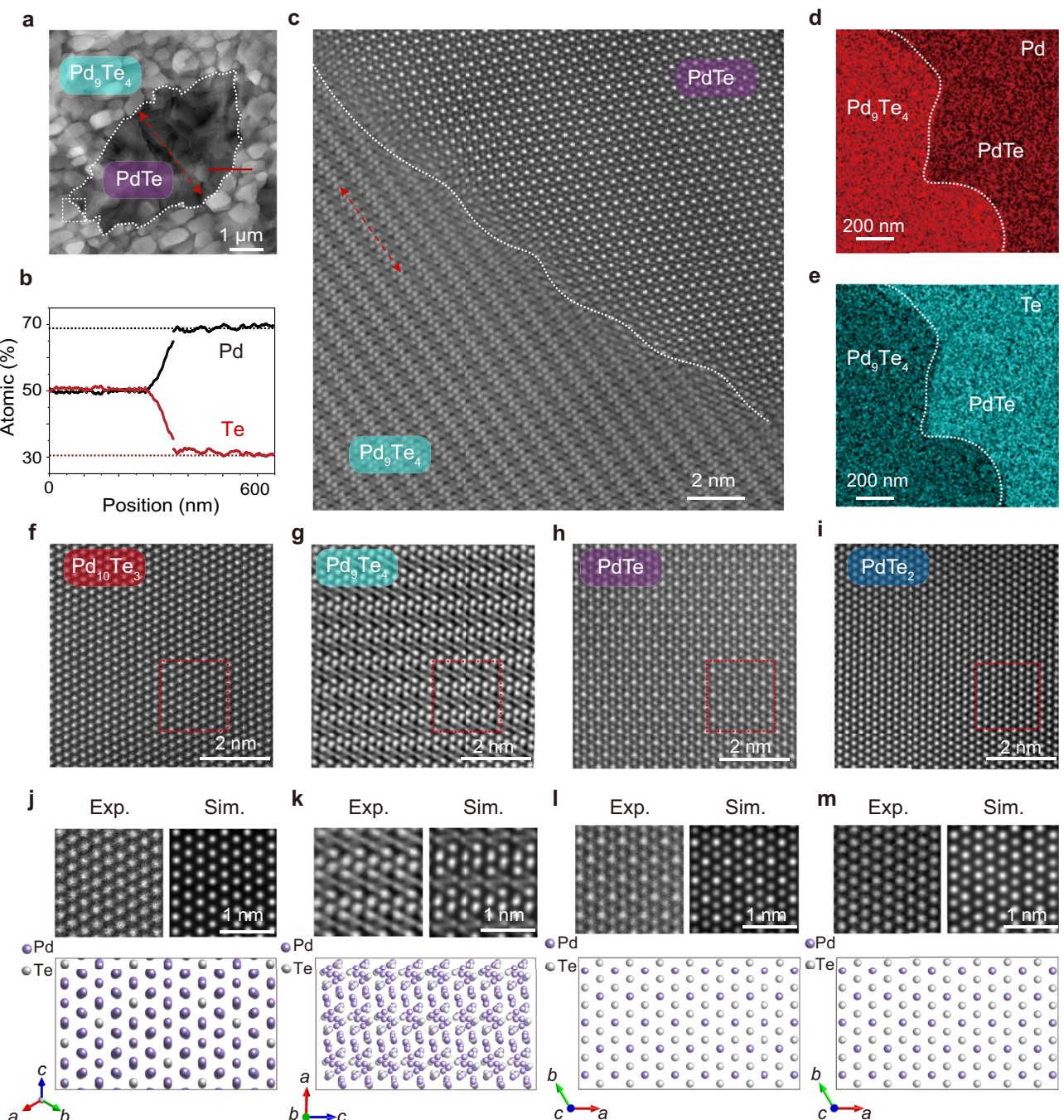

**Fig. 2 | Atomic-resolution STEM and EDX characterization. a** Low-magnification STEM image of PdTe nucleation, exhibiting elliptical features. **b** EDX line scan at the PdTe and $Pd_9Te_4$ interface. The molar ratio of Pd to Te changes from ~1:1 to 7:3. Source data are provided as a Source Data file. **c** Atomic-resolution STEM image at the PdTe and $Pd_9Te_4$ interface, the bright regions correspond to heavy atomic columns (Pd/Te clusters). The two phases seamlessly merge together without a distinct transition region. The long-chain direction of $Pd_9Te_4$ aligns with the short-axis direction of PdTe nucleation. **d–e** EDX scan (Pd: red; Te: green) at the PdTe and $Pd_9Te_4$ phase interface where color intensity scales with concentration, which shows a uniform color distribution on both sides of the interface but with significant intensity differences. **f–m** Atomic-resolution STEM images (experiment) of different nucleation sites and the corresponding simulation results confirm that the nucleation sites correspond to $Pd_{10}Te_3$, $Pd_9Te_4$, PdTe, and $PdTe_2$.

stoichiometry transition. Due to the low symmetry of $Pd_9Te_4$ lattice (monoclinic, $P2_1/c$), which has distinct crystallographic orientations and atomic arrangements that break the symmetry, it causes different interactions with the PdTe during the nucleation process. As a result, two different interfaces are formed with PdTe nucleation (hexagonal, $P6_3/mmc$) along the long (type I) and short axes (type II) (Supplementary Fig. 9), resulting in different phase transition rates and elliptical nucleation shapes. Interestingly, we also observe elongated PdTe

nucleation at some region (Supplementary Fig. 10). The cross-sectional STEM images show that the interface between PdTe and $Pd_9Te_4$ are also atomically sharp with seamless bonding, but the crystal orientation is different from that of the elliptical nucleation ($c$-axis is normal to the substrate), thus forming different interfaces and resulting in enhanced anisotropy of the growth rate (Supplementary Fig. 11). So interface plays an important role on the phase transition. The lattice anisotropy of the background and the nucleation results in different

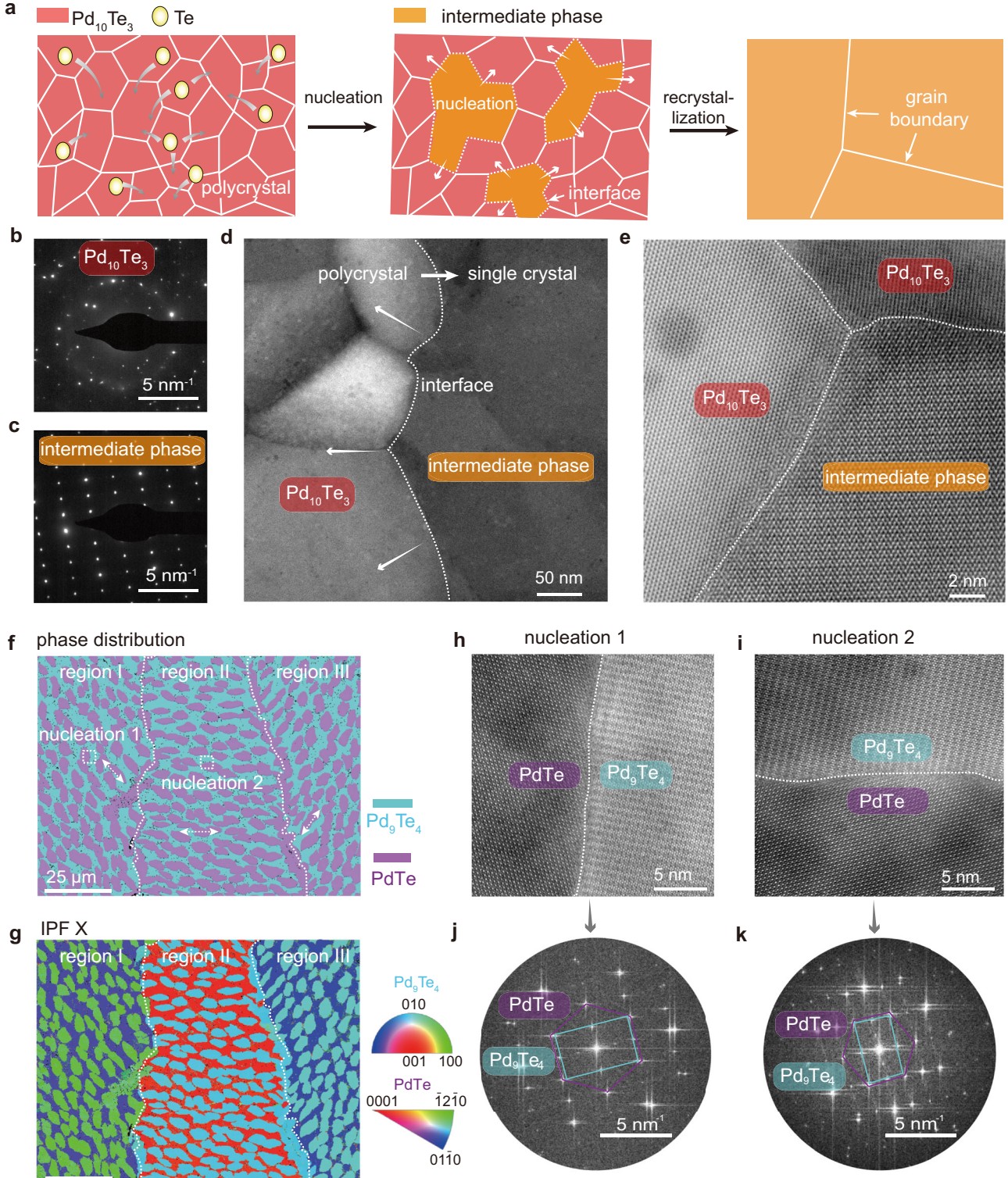

**Fig. 3 | Re-crystallization behavior and aligned nucleation. a** A schematic diagram of intermediate phase nucleation and growth from polycrystalline $Pd_{10}Te_3$ thin films. **b** The SAED pattern of the $Pd_{10}Te_3$ phase displays multiple sets of diffraction lattices (polycrystalline). The bright diffraction spots correspond to Bragg reflections from crystalline planes, with intensity proportional to the structure factor of the respective planes. **c** The SAED pattern of the intermediate phase exhibits a single set of rectangular lattice, indicating a single crystal structure within the region. **d** Low-magnification STEM image at the interface between $Pd_{10}Te_3$ and the intermediate phase, showing seamless integration of nanocrystals spanning tens of nanometers between $Pd_{10}Te_3$ and the intermediate phase. **e** Atomic-resolution STEM image at the interface between $Pd_{10}Te_3$ and the intermediate phase. **f, g** Phase distribution and corresponding IPF X-map of the PdTe nucleation region. Based on the orientation of the PdTe nucleations, this region can be divided into three parts labeled by the white dashed lines. **h, i** Atomic-resolution HAADF-STEM images of the interface between $Pd_9Te_4$ and PdTe at different nucleations marked by the white box in (**f**). **j** and **k**, the corresponding FFT images taken from (**h, i**) which show the same lattice orientation relationship.

interfaces, which in turn give rise to different interface energies and the relative growth rates of different facets, thereby causing the different shapes of different nucleations (Supplementary Figs. 7–9a). As $Pd_{10}Te_3$ (cubic, $Fd\bar{3}m$) is a polycrystal with a grain size of several tens of nanometers, the intermediate phase will encounter grains with various orientations during the phase transition process, thus forming different heterogeneous interfaces (Fig. 3a). These interfaces will have different growth rates, leading to the irregular shape of the intermediate phase nucleation. Co-planar heterostructures play a crucial role in specific fields, such as lowering the contact barrier[40,41], enhancing hydrogen evolution reaction[42] and creating quantum interface state[43,44], offering unique properties and functionalities that cannot be achieved by single-component materials. The sequential nucleation process offers a method for the preparation of various co-planar heterostructures. For example, we find a three-phase interface where the $Pd_9Te_4$ nucleation extend to the boundary of the intermediate phase, resulting in the formation of an intersection of the three phases of $Pd_{10}Te_3$, the intermediate phase, and $Pd_9Te_4$ (Supplementary Fig. 12). More complex co-planar and even vertical heterostructures can be created by carefully designing the nucleation process and vertical stoichiometry transition with the help of deposited Pd on surface[24].

## Kinetic mechanism of the stoichiometry transition

Apart from the changes in lattice structure, we further investigate the kinetic mechanisms of the stoichiometry transition associated with nucleation and growth. Direct tellurization of metal films typically yields films with nanometer size crystal grains[12,17], phase transition process and the accompanying re-crystallization provide unique method for improving the crystallinity of thin films (Fig. 3a). The SAED pattern of the $Pd_{10}Te_3$ phase shows multiple sets of diffraction spots (Fig. 3b), signifying the polycrystalline nature of the $Pd_{10}Te_3$ film. In the low-magnification STEM images, distinct crystal grains are visible with grain boundaries, and the grain size is in the tens of nanometers (Fig. 3d). At the interface between the $Pd_{10}Te_3$ and the intermediate phase (marked by white dashed lines), the differently oriented grains of the $Pd_{10}Te_3$ phase seamlessly connect to the intermediate phase (Fig. 3d, e).

Unlike the $Pd_{10}Te_3$ phase, the intermediate phase region exhibits uniform contrast without distinct grain boundaries, and its SAED pattern shows only a single set of square diffraction spots (Fig. 3c). The single-crystal nature of the intermediate phase nucleation is also confirmed by the atomic-resolution STEM images at different positions within a single nucleation, which show a completely consistent lattice orientation (Supplementary Fig. 13). This indicates that the stoichiometry transition process from $Pd_{10}Te_3$ to intermediated phase involves not only a change in lattice structure but also a re-crystallization process. At the interface, the atoms in the different grains of $Pd_{10}Te_3$ rearrange themselves using the lattice of the intermediate phase as a template. This re-crystallization behavior makes each individual nucleation a single crystal. By controlling the nucleation density and the rate of phase transition, we are able to regulate the crystalline quality of the thin films.

We then study the subsequent nucleation within the single-crystalline grains on a more macroscopic scale using electron back-scatter diffraction (EBSD). From the band contrast image, PdTe nucleations with ellipse shape are embedded in the $Pd_9Te_4$ background (Supplementary Fig. 14). According to the orientation of the PdTe nucleations, the entire region can be divided into three parts, with the interfaces between adjacent parts marked by white dashed lines. From the scanned phase distribution map, the entire region displays uniform purple and cyan colors, corresponding to the PdTe and $Pd_9Te_4$ phases (Fig. 3f). Within every single part, the contrast of the inverse pole figure scan along the $X$ axis (IPF X) of $Pd_9Te_4$ is consistent, indicating the same lattice orientation (Fig. 3g). The grain size of $Pd_9Te_4$ can reach tens of micrometers, which is three orders of

magnitude larger than that of the $Pd_{10}Te_3$ phase, further confirming the re-crystallization process involved in the phase transition. For different parts, the contrast varies, suggesting that these three parts originate from independent nucleation and their lattice orientations are random.

As for the PdTe nucleation, their lattice orientations change along with the $Pd_9Te_4$ background, but they are consistent within the same part (Fig. 3g and see IPF Y in Supplementary Fig. 14b). To clarify this, we obtained the atomic-resolved STEM images of the interface between the two PdTe nucleations and $Pd_9Te_4$ (Fig. 3h–i). Although the HAADF-STEM images reveal the different lattice orientations of the two PdTe nucleations and the $Pd_9Te_4$ background, the fast Fourier transform patterns suggest a fixed orientation relationship between them (Fig. 3j–k). This fixed lattice orientation relationship leads to the orientation of PdTe nucleation being completely determined by that of the $Pd_9Te_4$ background. The IPF scans along the $Z$-axis (perpendicular to the substrate direction) in three distinct regions demonstrate a consistent crystallographic orientation (Supplementary Fig. 14c), indicating that the (010) and (0001) planes of the two phases are with the lowest surface energy among various crystal planes.

## Stoichiometry-controllable wafer-scale growth

Stoichiometry-controllable wafer-scale growth is the prerequisite for achieving device integration and application. However, preferential nucleation at the substrate edge leads to the acquisition of only mixed phases. To address this issue, we need to realize: (1) spatially homogeneous nucleation and (2) the ability to control the phase that is ultimately formed. To achieve spatially uniform nucleation and growth, we use another substrate coated with a Te film, placed face-to-face with the Pd substrate to uniformly supply Te (Fig. 4b). By depositing onto a transparent quartz substrate, we are able to capture the growth process through the coating layer. A homogeneous nucleation process is observed by using in-situ CVD system (see Supplementary Movie 3). The nucleation process provides us with barriers to regulate the phase that is ultimately formed. By precisely manipulating the thickness of the Te film, we are capable of regulating the amount of Te involved in the reaction. As the stoichiometry transition takes place in the direction of increasing Te content, we can thereby control the cessation of the phase transition, and as a result, enable stoichiometry-controllable growth. The obtained films are continuous, smooth, and uniform over a large scale (Supplementary Fig. 15). From the $Pd_{10}Te_3$ phase to the $PdTe_2$ phase, the thickness of the film increases from 14.1 nm to 23.2 nm, and the Ra value slightly rises from 0.35 nm to 1.3 nm. We characterized the phase homogeneity of PdTe and $PdTe_2$ using Raman spectroscopy and XRD. The Raman peaks at different positions within the wafer are uniform, indicating that the obtained thin films are of a pure phase (Supplementary Fig. 16). The (0 0 l) peaks indicate that the natural surface is the a-b plane, which is consistent with the EBSD and STEM results (Supplementary Fig. 17). As for the $Pd_9Te_4$ and $Pd_{10}Te_3$ phases, we obtained the EBSD patterns and atomic structures at different positions of the wafer, all of which showed consistency (Supplementary Figs. 18–19).

Therefore, we conduct CVD growth for Pd films ranging from 5 nm to 20 nm with a step of 5 nm, while gradually increasing the thickness of the Te film until the resulting phases are pure $PdTe_2$. For Pd films with thickness exceeding 10 nm, we can sequentially obtain pure $Pd_{10}Te_3$, $Pd_9Te_4$, PdTe, and $PdTe_2$ phases at the appropriate thickness of Te (Fig. 4b). If the thickness of the Te film is between these specific thicknesses, the corresponding mixed phases will be obtained. Due to the narrow nucleation window between the intermediate phase and the $Pd_9Te_4$ phase, the nucleation of the $Pd_9Te_4$ phase begins before the complete coalescence of the intermediate phase, making us obtain a film with a three-phase mixture of $Pd_{10}Te_3$, intermediate and $Pd_9Te_4$ phase (see Supplementary Movie 4). By coordinating the thicknesses of Pd and Te films, we eventually obtain a phase diagram

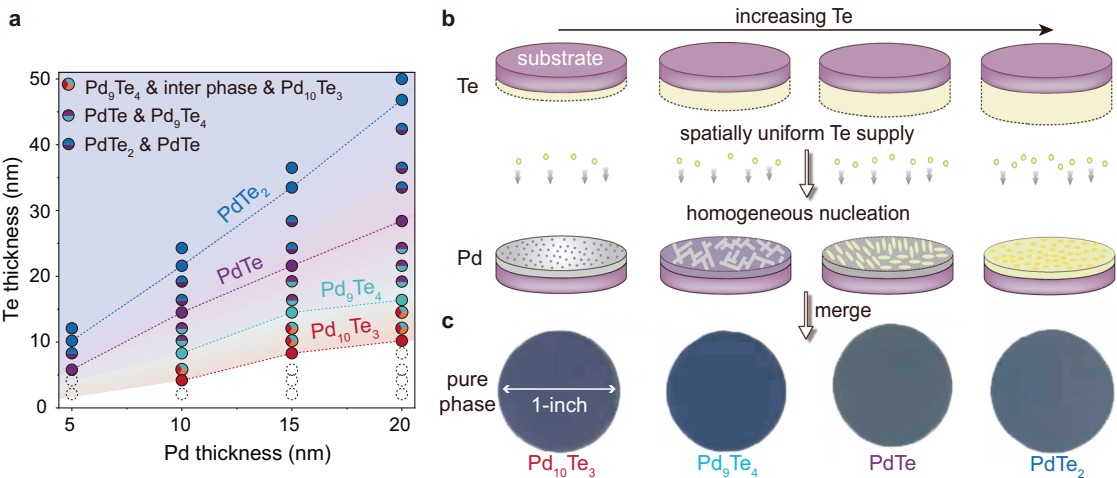

**Fig. 4 | Stoichiometry-controllable wafer-scale growth. a** The phase diagram of the Pd-Te compound shows that different stoichiometries can be obtained under different thicknesses of Pd and Te. Source data are provided as a Source Data file. **b** Schematic diagram illustrating the stoichiometry-controllable synthesis of wafer-scale Pd-Te films. **c** The optical images of the 2.5-cm (1-inch) $Pd_{10}Te_3$. $Pd_9Te_4$. PdTe and $PdTe_2$ wafers.

(Fig. 4a). According to the phase diagram, we successfully achieve the growth of 1-inch $Pd_{10}Te_3$, $Pd_9Te_4$, PdTe, and $PdTe_2$ wafers (Fig. 4c). We note that the maximum size of the wafers is solely determined by size of the furnace, not by technical limitations. Because the phase transition occurs through co-planar interfaces, it is not affected by the lattice constant and symmetry of the substrate, allowing us to integrate different phases of Pd-Te thin films on arbitrary substrates, including amorphous $SiO_2$, single crystalline Si, and sapphire (Supplementary Fig. 20).

## Superconducting properties

$PdTe_2$ has been confirmed experimentally a topological Dirac semi-metal with superconductivity and the superconducting transition has also been observed in some other $PdTe_x$ compounds[45–67]. However, the direct correlation between their superconducting properties and specific phases remains unclear. This ambiguity hinders the investigation of the physical properties and the application of devices based on the Pd-Te films. Controllable phase preparation offers an opportunity to address this issue. We study the superconducting properties of different phases in the two-dimensional (2D) limit based on the synthesized high-crystalline $Pd_{10}Te_3$, $Pd_9Te_4$, PdTe and $PdTe_2$ films. By adopting the four-terminal method, measurements of temperature-dependent resistance (the $R–T$ curve) in different phases are carried out. The obtained $R–T$ curves show a sudden drop at -1.1 K, 0.31 K, 4.5 K, and 1.66 K, with the resistance reaching zero at -1.04 K, 0.18 K, 3.8 K, and 1.52 K, implying that the samples undergo a superconducting phase transition (Fig. 5a–d). The critical temperature $T_c$ for the superconductivity, which is defined as the temperature at which $R = 90\% \times R_0$ ($R_0$ is the normal state resistance), is estimated to be 1.09 K, 0.29 K, 4.45 K, and 1.65 K. The $T_c$ of PdTe and $PdTe_2$ are consistent with that of the exfoliated materials[24,37]. The PdTe device retained its superconducting properties, with scarcely any notable alteration in the superconducting transition temperature even after being exposed to air for a period of one year (Supplementary Fig. 21). This additional evidence substantiates the favorable stability of the as-grown PdTe thin films.

The $T_c$ of these four phases do not exhibit a monotonicity as the Te component increasing, among which PdTe has the highest transition temperature. We performed the first-principle calculations to obtain the density of states DOS values of the four phases. According to the Bardeen-Cooper-Schrieffer (BCS) theory, the superconducting transition temperature is closely related to the electronic density of states at the Fermi level. A higher density of states means that there are

more electronic states near the Fermi level available for electrons to form Cooper pairs, which is conducive to the occurrence of super-conductivity. In our case, the PdTe has the largest density of states at the Fermi level (Supplementary Fig. 22), thus having the highest superconducting transition temperature[37]. The density of states of other phases decreases in the order of $PdTe_2$, $Pd_{10}Te_3$, and $Pd_9Te_4$ (Supplementary Fig. 22), which is consistent with the superconducting transition temperature. We then investigate the dimensionality of the superconductivity of the different Pd-Te phases by performing the temperature dependence measurement of resistance under out-of-plane and in-plane magnetic fields (Figs. 5a–d and Supplementary Fig. 23). A positive magneto-resistance (MR) is observed with the transition temperature shifting gradually toward lower temperature for both the in-plane and out-of-plane magnetic fields. The critical field in an in-plane field ($H_{//}$) is enhanced compared with that in perpendicular field ($H_\perp$), indicating the 2D nature of the superconductivity (Supplementary Fig. 24).

In summary, we verified the feasibility of applying stoichiometry as a degree of freedom to phase engineering. In total, we identified five phases featuring different stoichiometric ratios in the Pd-Te binary compound. By regulating the atomic diffusion process through temperature and combining with the in-situ CVD system, we reveal a series of kinetic processes of stoichiometry transitions. These stoichiometry transitions occurred in the form of various types of nucleation. We also confirm the re-crystallization process from $Pd_{10}Te_3$ to the intermediate phase and a fixed lattice orientation relationship in the subsequent nucleation, offering method for enhancing the crystallization quality of the films. Through controlling spatially homogeneous nucleation and the termination of the phase transition, we achieve stoichiometry-controllable wafer-scale growth. Eventually, we examine the super-conducting behaviors and affirm the direct correlation between their superconductivity and specific phases. Our work offers notable approaches and insights for enhancing the diversity of phase engineering and scale-up application in nanomaterials

## Methods

### In-situ CVD growth

The Pd-Te films are synthesized by tellurizing the Pd film at atmospheric pressure using a commercial horizontal hot-wall tube furnace equipped with optical microscopy (Unit Micro-STS1200). Pd film of 10 nm is deposited on Si/$SiO_2$ substrates through thermal evaporation. The furnace is ramped to 300 °C in 5 minutes and is kept at the temperature for 30–60 min with 5% $H_2$/Ar mixed gas flow at rates of 20

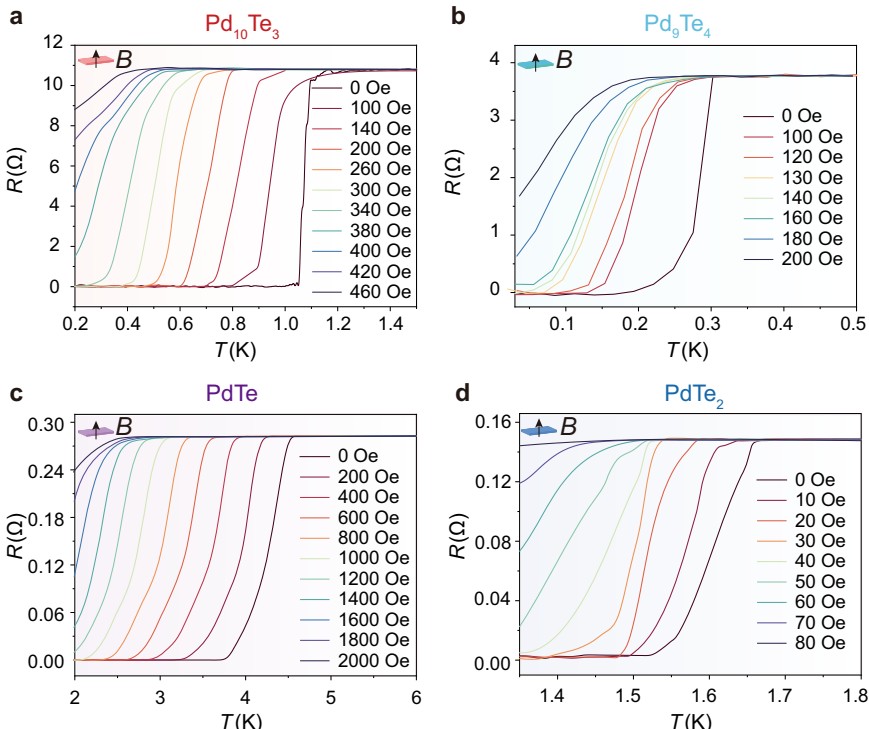

**Fig. 5 | Superconducting characteristics of Pd$_{10}$Te$_3$, Pd$_9$Te$_4$, PdTe and PdTe$_2$.**
**a–d** Temperature dependence of the resistance ($R$–$T$ plots) of the Pd$_{10}$Te$_3$, Pd$_9$Te$_4$, PdTe and PdTe$_2$ at different out-of-plane magnetic fields. At 0 Oe, a sudden resistance drops at -1.1 K, 0.31 K, 4.5 K, and 1.66 K is observed, implying that the samples undergo a superconducting phase transition. Source data are provided as a Source Data file.

standard cubic centimeters per minute (sccm). Record the growth process through the quartz window with an optical microscope.

**Stoichiometry-selected synthesis of the wafer-scale Pd-Te film**
Pd film of 10 nm is deposited on Si/SiO$_2$, Si or sapphire substrates through thermal evaporation. Then, 4.2 nm, 8.3 nm, 14.5 nm and 21.6 nm Te thin films are evaporated on the SiO$_2$ substrate, which are respectively used for growing Pd$_{10}$Te$_3$, Pd$_9$Te$_4$, PdTe and PdTe$_2$. The Te thin films are placed face to face to cover the Pd substrate. The substrates are put into a 2-inch diameter tube furnace. The furnace is ramped to 300 °C in 10 minutes and is kept at the temperature for 30 min with 20 sccm 5% H$_2$/Ar mixed gas. After that, the furnace is naturally cooled down to room temperature.

**Simulation of the Pd-Te phases**
HAADF image simulations were obtained using the STEM_CELL software simulation package matching the microscope experimental settings (acceleration voltage of 300 kV; convergence semiangle of 25 mrad; detector collection semiangle of 80 – 379 mrad) and using supercells with -10 nm thickness.

**Transport measurement**
The electronic transport measurements were performed in a dilution refrigerator (Oxford Triton 500) with temperatures from 30 mK to 6 K. The resistance of the sample was measured by the standard four-probe method using Keithley 6221/2182 A Delta Mode with an excitation current of 1 mA.

**First-principle calculations**
Our density functional theory calculations were performed using the projector augmented wave method as implemented in the Vienna ab initio simulation package (VASP). The exchange-correlation functional for describing the electron interactions was generalized gradient approximation with Perdew-Burke-Ernzerh of parametrization. The

uniform Γ-centered k mesh of $12 \times 12 \times 6$, $12 \times 12 \times 6$, $4 \times 2 \times 4$ and $3 \times 3 \times$ was adopted for integration over the Brillouin zone (BZ) for PdTe, PdTe$_2$, Pd$_9$Te$_3$ and Pd$_{10}$Te$_4$, respectively. The cutoff energy of 500 eV was set for the plane wave basis. During the geometrical optimizations process, the convergence criteria of energy and force of the geometrical optimizations were set to $10^{-7}$ eV and 1 meV/Å per atom.

**Characterization**
The EBSD characterizations are carried out using Tescan mira 3 LMH SEM equipped with the Oxford instrument symmetry detector. The STEM characterizations are carried out using a Thermo Fisher Scientific Titan Themis Z 60–300 kV Electron Microscope with condenser lens and objective lens aberration correctors; EDS mapping are carried out through Bruker Super-X EDX detector. In order to acquire better atomic images, this instrument was operated at an atomic resolution of 80 pm with a 30 pA beam, convergence semiangle of 25 mrad, and collection semiangle snap of 80 – 379 mrad at 300 kV. All selected area electron diffraction (SAED) patterns are carried out under a 10 μm aperture aperture at 300 kV.

## Data availability
Source data are provided with this paper. The crystallographic data used in this study were obtained from the Material Project database, and detailed information can be found in the Supplementary Information. Additional data is available from the corresponding author upon request Source data are provided with this paper.

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

## Acknowledgements

The authors acknowledge Analysis & Testing Center, Beijing Institute of Technology for the use of STEM. This work is supported by the National Key R&D Program of China (Grants Nos. 2023YFB3611700, X.X.; 2022YFA1403700, C.L.), the National Natural Science Foundation of China (Grants Nos. 62274010, X.X.; 92163206, Y.W.; 12321004, Y.W.; 12374036, C.L.; 12274025, R.S.), Young Elite Scientists Sponsorship Program by CAST (Grants Nos. 2023QNRC001, X.X.), the open research fund of Suzhou Laboratory (SZLAB-1508-2024-TS014, X.X.) and the Fundamental Research Funds for the Central Universities (2024CX06086, M.H.).

## Author contributions

Y.W. and X.X. supervised the project. X.X. conceived the idea. X.X. and M.H. designed the experiments and analyzed the results. Z.H. performed the STEM characteristics under the supervision of R.S. R.G. performed the STEM simulation under the supervision of W.Z. Z.R. conducted the transport measurements under the supervision of C.L. H.C., D.Z., and Y.D. help to set up the CVD system under the supervision of X.X., Y.H. and Y.W. S.Y. and P.G. help with the Raman measurement and transport data analysis, respectively, under the supervision of Y.Y. X.X. wrote the manuscript with input from all the authors. Y.W. reviewed and edited the manuscript. All the authors help with the discussion.

## Competing interests

The authors declare no competing interests.
