## [Transparent Peer Review file · Nature Communications]

Stoichiometry-engineered phase transition in a two-dimensional binary compound

Corresponding Author: Professor Xiaolong Xu

Version 0:

Reviewer comments:

Reviewer #1

(Remarks to the Author)

This manuscript reports the observation of various Pd-Te binary phases during the formation of PdTe₂ through the annealing of Pd films in a Te atmosphere, demonstrating the growth of Pd-Te binary films with different stoichiometries. The main idea of the study is innovative, and the results are novel, so I believe that with the following improvements, this work could be published in Nature Communications.

1. The authors mention lowering the growth temperature to 300 °C, below the melting temperature of Te, to reduce the diffusion rate. In this case, Te vapor could condense on the quartz tubes or substrates. Was this issue observed during the process?
2. The overall process needs to be more clearly divided, such as into (1) the initial tellurization of the Pd film to form Pd₁₀Te₃, and (2) the stoichiometric phase transition in the Pd-Te system. For Pd₁₀Te₃, nucleation occurs at the edges and propagates toward the center, while the stoichiometric phase transition seems to involve simultaneous multi-nucleation and merging, as shown in Fig. 3a.
3. The TEM results suggest that Pd₉Te₄, Pd₁₀Te₃, and the intermediate phases exhibit periodic lattice structures. Therefore, it would be interesting to see other results, such as Raman spectra with a different x-axis range.
4. The large-area film growth results are shown only in photos. It is necessary to more convincingly demonstrate the uniform large-area growth of each phase through XRD and AFM.
5. I am curious whether this technique (or mechanism) is specific to the Pd-Te system or if it can be applied to other metals or chalcogens as well.

Reviewer #2

(Remarks to the Author)

Manuscript No. NCOMMS-24-63178-T

Authors: M. Huang et al.,

Title: Stoichiometry-engineered phase transition in a two-dimensional binary alloy

The above manuscript reports the mixing of Te and Pd when they are face to face at 300 °C. The authors analyze the product at different reaction time using EDX, STEM, SAED, and EBSD. They find that there is stoichiometric transition from Pd₁₀Te₃ to an intermediate phase to Pd₉Te₄ to PdTe to PdTe₂. However, I see neither science nor engineering. In the following, we provide more detailed comments.

1. With phases reported in the manuscript, I do not see evidence of "alloy" as Pd_yTe_x the authors reported are actually compounds with crystal structures that do not resemble to either Pd or Te. In addition, all compounds (Pd_yTe_x) were not engineered but rather a diffusion process. Therefore, I believe that the title is misleading.

2. Identifying five PdyTex phases does not mean that they are purposely engineered. Instead, it implies that different PdyTex phase requires different reaction conditions.

3. References are incorrectly cited. For example, references 41-43 have nothing to do with PdTex. On the other hand, literature related to PdTe and PdTe₂ are not cited.

Reviewer #3

(Remarks to the Author)

The manuscript titled "Stoichiometry-engineered phase transition in a two-dimensional binary alloy" introduces stoichiometry as a novel approach to phase engineering in Pd-Te binary alloy, enabling controlled wafer-scale growth through reduced diffusion rates and precise nucleation. This study identifies five distinct phases, including four with superconducting properties, offering valuable insights into phase transition and advancing scalable applications in nanomaterials.

I find this manuscript to be a well-presented study. The authors have clearly articulated their findings in a well-structured organization. The manuscript provides sufficient background information, along with a detailed and comprehensive description of the methodology used. This attention to clarity and detail enhances the readability and reproducibility of the research. However, I suggest a minor revision to address the following points:

1. The manuscript emphasizes the role of the interface in driving phase transitions and influencing nucleation shapes. However, the discussion lacks depth regarding the impact of vertical phase heterogeneity. For example, in Figure 3d, the irregular shape of the intermediate phase is attributed to the polycrystallinity of the Pd₁₀Te₃ background, yet this explanation is not sufficiently detailed. The authors should provide a more thorough analysis of this phenomenon and propose strategies to mitigate vertical heterogeneity. Addressing this issue would be critical for enhancing phase uniformity and the overall quality of the material.

2. Regarding the vertical phase heterogeneity mentioned earlier, the authors claim to achieve pure single phases by controlling the Pd and Te thickness. However, while optical microscopy (OM) observations (Figure 4) and video evidence are provided, these alone may not be sufficient to confirm a single phase across the entire wafer. I suggest the authors include XRD measurements to verify this claim.

3. The manuscript lacks sufficient detail on phase stability, which is crucial for practical applications. The authors should elaborate on the thermodynamic and operational stability of each phase to enhance the study's relevance and applicability.

4. The manuscript mentions that the T_c of the four superconducting phases does not exhibit a monotonic trend, but it does not provide sufficient explanation for this behavior. Additionally, while PdTe is highlighted as having the highest density of states (DOS), contributing to its higher T_c, the DOS values for the other phases are not discussed. Including these values and analyzing their relationship with T_c could provide valuable insights into the observed trend and improve the overall depth of the study.

Reviewer #4

(Remarks to the Author)

Version 1:

Reviewer comments:

Reviewer #1

(Remarks to the Author)

The author had adequately addressed all of my questions and comments. It is particularly remarkable that a similar mechanism observed in the Mo-Te system. I believe this manuscript is suitable for publication in Nature Communications without any further revisions.

Reviewer #2

(Remarks to the Author)

Manuscript No. NCOMMS-24-63178A

Authors: M. Huang et al.,

Title: Stoichiometry-engineered phase transition in a two-dimensional binary compound

The revised manuscript has clarified some aspects of concerns. However, the overall story is about the formation of five phases. It still lacks basic information. For example, while Figure 1 shows the phase distribution through the process, these phases are later characterized via EDX, STEM, SAED, EBSD, and simulation. However, there is no information about the crystal structure and corresponding lattice parameters of each phase. The lack of such information makes readers difficult to follow the description. In view of Fig. 1(f), this reviewer cannot tell the difference between spectra for three phases. In

line#203, the authors state that "Due to the low symmetry of Pd₉Te₄ lattice, two different interfaces are formed with PdTe...". Without knowing the symmetry of Pd₉Te₄ and PdTe, one cannot follow the discussion. The authors should be aware that similar losing description occurs in multiple places.

Regarding the observation of superconductivity, the authors should be aware that superconductivity has been heavily studied in bulk PdTe and PdTe₂. The relevant references have not been included even in the revised manuscript.

Reviewer #3

(Remarks to the Author)

I appreciate the authors' efforts to address the comments from the first round of review. The revisions have significantly improved the manuscript, and most of my concerns have been satisfactorily addressed. Below, I provide my detailed feedback on the revisions:

1. Vertical Phase Heterogeneity:

The authors have provided TEM and cross-sectional STEM images, showing the absence of Moiré fringes, which confirms vertical phase homogeneity. The observed fringes at the interface (~2 nm) are well-justified. The data sufficiently addresses my concerns regarding vertical heterogeneity.

2. Confirmation of Single Phase Across Wafer:

The authors followed my suggestion by including XRD and Raman data for PdTe and PdTe₂, as well as EBSD for Pd₉Te and STEM for Pd₁₀Te₃. This comprehensive data set confirms the presence of distinct phases across the wafer. However, I am curious whether the coexistence of multiple phases on the same wafer induces strain. Could the authors comment on whether strain arises in the mixed-phase samples due to the expansion during tellurization?

3. Phase Stability:

The authors have demonstrated good thermodynamic stability by annealing the samples at 500°C without Te, with no significant changes observed in OM and Raman data. Additionally, the superconductivity test on a device stored for one year confirms long-term stability. However, I would like to know at what temperature the stability is lost. Since Te typically degrades around 550°C, will the compound undergo phase transitions above this temperature?

4. Density of States (DOS):

The authors have provided DOS data for all phases, fully addressing my concerns. I have no further comments on this point.

In summary, the authors have made substantial improvements to the manuscript, and the additional data provided are sufficient to address most of my initial concerns. I recommend minor revisions to address the remaining points regarding strain and phase stability at higher temperatures.

Reviewer #4

(Remarks to the Author)

Version 2:

Reviewer comments:

Reviewer #2

(Remarks to the Author)

The authors have addressed my concerns/comments and the 2nd revised manuscript is more clear. I now recommend the publication of the 2nd revised manuscript.

Reviewer #3

(Remarks to the Author)

Overall, I believe the authors have adequately addressed all the questions from the previous round, and no further revisions are needed.

Reviewer #4

(Remarks to the Author)

Responses to Reviewer #1

General comments: *“This manuscript reports the observation of various Pd-Te binary phases during the formation of PdTe₂ through the annealing of Pd films in a Te atmosphere, demonstrating the growth of Pd-Te binary films with different stoichiometries. The main idea of the study is innovative, and the results are novel, so I believe that with the following improvements, this work could be published in Nature Communications.”*

Response: We greatly appreciate Reviewer #1's precise summary of our work and the positive evaluation. We are encouraged that the reviewer found that *“The main idea of the study is innovative, and the results are novel”* and *“believe that with the following improvements, this work could be published in Nature Communications”*. His/her comments/suggestions are very helpful in improving the quality of our manuscript. Below, we provide a point-by-point response to his/her remaining points, with the corresponding changes highlighted in the revised manuscript and/or Supplementary Information.

Comment: *“The authors mention lowering the growth temperature to 300 °C, below the melting temperature of Te, to reduce the diffusion rate. In this case, Te vapor could condense on the quartz tubes or substrates. Was this issue observed during the process?”*

Response: We thank the reviewer for raising this detailed concern. To investigate the condensation of Te on quartz tubes, we carried out experiments using a new quartz tube with transparent tube walls (Fig. R1a). After growing at 300 °C for 30 minutes in the presence of Te, no significant changes were observed on the quartz tube walls (Fig. R1b). Subsequently, when the growth time was extended to 300 minutes, a light gray contrast appeared on the tube walls, indicating a small amount of Te deposition. Considering such a long growth time, it implies that the sublimation rate of Te at this temperature is extremely slow, enabling us to observe the phase transition processes of different stoichiometric ratios.

Fig. R1 The condition of the tube. (a) a new quartz tube. (b) the quartz tube after a 30-minute experiment at 300°C. (c) the quartz tube after total 300-minute experiment at 300°C.

We then utilized STEM and EDX to study the condensation of Te on the substrate. Taking the Pd₁₀Te₃ thin film shown in Fig. R2 as an example. Through EDX mapping of the surface of the Pd₁₀Te₃ thin film, we found that the distributions of Te and Pd elements were uniform across the film surface. The molar ratio of Pd to Te elements was approximately 10:3, which was basically consistent with the stoichiometric ratio of Pd₁₀Te₃, suggesting that there was no additional Te deposition on the film surface.

Fig. R2 A large-area STEM-EDX analysis of Pd₁₀Te₃ to study the condensation of Te on the substrate. (a) Low-magnification STEM image of Pd₁₀Te₃. (b) EDX mapping of Pd elements. (c) EDX mapping of Te elements. (d) The elemental composition and ratio of Pd₁₀Te₃. The ratio of Pd and Te in the analyzed region corresponds to the atomic ratio of Pd₁₀Te₃, confirming that no additional Te is present.

To further confirm whether there was Te deposition on the film surface, we performed cross-sectional STEM characterization on the PdTe-Pd₉Te₄ thin film shown in Fig. R3, and found that the entire film transformed into PdTe or Pd₉Te₄ without additional Te

deposition on the surface. The above results collectively demonstrate that the Te always reacted with Pd films, without the precipitation of elemental Te.

Fig. R3 The interface STEM characterization in order to confirm the smooth surface without detectable Te particles. **a**, Low-magnification STEM image of PdTe thin film. **b**, Atomic-resolution HAADF-STEM image of PdTe thin film.

Comment: “*The overall process needs to be more clearly divided, such as into (1) the initial tellurization of the Pd film to form Pd₁₀Te₃, and (2) the stoichiometric phase transition in the Pd-Te system. For Pd₁₀Te₃, nucleation occurs at the edges and propagates toward the center, while the stoichiometric phase transition seems to involve simultaneous multi-nucleation and merging, as shown in Fig. 3a.*”

Response: We appreciated the reviewer for raising this constructive suggestion. As the reviewer pointed out, the entire phase transition proceeds in steps. The initial tellurization of the Pd film to form Pd₁₀Te₃ occurs at the edges, followed by the stoichiometric phase transition in the Pd-Te system. Following the reviewer's suggestion, in Fig R4 (Fig. 1a-b in revised manuscript), we provided a clear description of step1 and step2: (1) the initial tellurization of the Pd film to form Pd₁₀Te₃, and (2) the stoichiometric phase transition in the Pd-Te system. In summary, the annotation of the simultaneous multi nucleation and merging process were added in the revised Fig. 1b in the main text, and the detailed descriptions were also provided in the revised manuscript in page 3.

Fig. R4 Multi-step stoichiometric transition and diverse nucleation processes. (a) Schematic diagram of the tellurization process of Pd films and the stoichiometric phase transition. (b) Optical image of the intermediate state of multi-nucleation and merging.

Comment: “The TEM results suggest that Pd₉Te₄, Pd₁₀Te₃, and the intermediate phases exhibit periodic lattice structures. Therefore, it would be interesting to see other results, such as Raman spectra with a different x-axis range.”

Response: We appreciated the reviewer for raising this constructive suggestion. Following the reviewer's suggestion, we conducted Raman spectra measurement on the Pd₁₀Te₃ phase, intermediate phase, and Pd₉Te₄ phase within the range from 50 cm⁻¹ to 2000 cm⁻¹ as shown in Fig. R5. However, apart from the silicon Raman peak of the substrate, no distinct Raman peaks were detected. To avoid the omission of Raman peaks due to resolution, we also carried out measurements using a grating with 2400 g/mm. Nevertheless, no significant Raman peak was obtained. Since both Pd and Te elements are relatively heavy, we believe that the Raman signals of these phases will not exceed 2000 cm⁻¹. The cross section of Raman scattering is so low that usually only one Raman photon is produced for every 10⁶-10¹⁰ incident photons for different materials (Overview of Raman Spectroscopy: Fundamental to Applications. *Modern Techniques of Spectroscopy*, Springer, 13, 145–184, 2021). The cross section of Raman scattering of these three phases might be too small, leading to a relatively weak Raman signals. Special experimental techniques, such as tip enhanced Raman scattering (TERS)

spectroscopy or infrared spectroscopy measurements, are required for further measurement to help us get more information.

Fig. R5 The Raman spectra of Pd₁₀Te₃ at (a) 600 mm⁻¹ and (b) 2400 mm⁻¹. The Raman spectra of intermediate phases at (c) 600 mm⁻¹ and (d) 2400 mm⁻¹. The Raman spectra of Pd₉Te₄ at (e) 600 mm⁻¹ and (f) 2400 mm⁻¹. No obvious Raman peaks are observed.

Comment: “The large-area film growth results are shown only in photos. It is necessary to more convincingly demonstrate the uniform large-area growth of each phase through XRD and AFM.”

Response: We appreciate the constructive feedback provided by the reviewer. Following the reviewer's suggestion, we performed new characterizations, including AFM, Raman, and XRD for the wafers of each phase.

Firstly, we used adhesive tape to peel off part of the Pd-Te thin film from the substrate, thus creating some steps to facilitate the measurement of the film thickness. As shown

in Fig. R6, we found that as the tellurization proceeded from $\text{Pd}_{10}\text{Te}_3$ to PdTe_2 , compared with the 10 nm Pd thin film, the thickness of the thin film gradually increased from about 14.1 nm to around 23.2 nm, and the roughness R_a gradually increased from 0.35 nm to about 1.3 nm. The thickness variation at different positions of each phase was relatively consistent, and the surface was relatively flat with a low roughness, indicating the uniformity of the large-area films.

Fig. R6 AFM height images and corresponding curve of $\text{Pd}_{10}\text{Te}_3$, Pd_9Te_4 , PdTe and PdTe_2 wafer measured at different positions respectively, indicating that the thickness of the film has good uniformity over a large scale. (a-d) $\text{Pd}_{10}\text{Te}_3$, (e-h) Pd_9Te_4 , (i-l) PdTe , (m-p) PdTe_2 .

Then, the homogeneity within a large range of each phase was verified by the X-ray diffraction (XRD) using Cu K α radiation, as shown in Fig. R7. The (0 0 1) peaks of PdTe₂ and PdTe phases indicate that the natural surface is the *a-b* plane, which is consistent with our EBSD and STEM results.

Fig. R7 The XRD spectra of (a) PdTe₂ and (b) PdTe phases.

However, for the Pd₉Te₄ and Pd₁₀Te₃ phases, no obvious diffraction peaks were observed, probably due to the thin film thickness and the small grain sizes. Therefore, we firstly characterized different positions of the Pd₉Te₄ wafer using EBSD (Fig. R8). Clear Kikuchi pattern diffractions could be seen at different positions, and they could be well fitted with the lattice of Pd₉Te₄, indicating that the films at different positions had good crystallinity and uniformity.

Fig. R8 Kikuchi pattern diffractions of the Pd₉Te₄ wafer using EBSD. (a) Optical images of

Pd₉Te₄ wafer. (b) Clear Kikuchi pattern diffractions could be seen at different positions, indicating that the films at different positions had good crystallinity.

Since the grain size of Pd₁₀Te₃ was too small, we performed STEM characterization of the Pd₁₀Te₃ wafer (Fig. R9). A distinct polycrystalline structure could be seen, with the grain size being in the range of tens of nanometers. When a single grain was magnified, a clear atomic structure that matched well with Pd₁₀Te₃ could be observed.

Fig. R9 STEM characterization Pd₁₀Te₃ films. (a) Low-magnification STEM image of polycrystalline Pd₁₀Te₃. (b-f) Atomic-resolution HAADF-STEM images of Pd₁₀Te₃ at different position.

We also characterized the Raman spectra of the grown PdTe and PdTe₂ wafers as shown in Fig. R10, and the Raman peak position and intensity at different positions were uniform. All of the above characterization results indicate that the large-area Pd-Te films we grew possess favorable uniformity.

In summary, the characterizations related to the uniformity and homogeneity of the synthesized wafers of each phase have been added to the revised Supplementary Information (Fig. S5-S9). Corresponding discussions have also been incorporated into the revised manuscript with highlights in page 10.

Fig. R10 The Raman spectra of PdTe and PdTe₂ wafer. (a-b) optical micrograph of PdTe wafer and the corresponding Raman spectra. (c-d) optical micrograph of PdTe₂ wafer and the corresponding Raman spectra. The Raman characteristic peaks indicate the film has good uniformity over a large scale.

Comment: “I am curious whether this technique (or mechanism) is specific to the Pd-Te system or if it can be applied to other metals or chalcogens as well.”

Response: We thank the reviewer for raising this open question. The concept of achieving controllable phase transformation by engineering the stoichiometric ratio can be applied to other material systems. However, different material systems require specific experimental designs. Here, we present some of our ongoing experimental results in the Mo-Te system. Unlike the Pd-Te system, in this system, we first need to obtain 1T'-MoTe₂, and then through precise control of the generation rate and concentration of Te vacancies in the thin film, the preparation of thin films with various stoichiometric ratios can be realized. Currently, we have achieved the controllable preparation of 1T'-MoTe₂, Mo₅Te₈, Mo₃Te₄, and Mo₆Te₆ phases [REDACTED]. The relevant data is being collated and we plan to submit it for publication in the near future.

[REDACTED]

[REDACTED]

[REDACTED]

[REDACTED]

Responses to Reviewer #2

General comments: *“The above manuscript reports the mixing of Te and Pd when they are face to face at 300 °C. The authors analyze the product at different reaction time using EDX, STEM, SAED, and EBSD. They find that there is stoichiometric transition from Pd₁₀Te₃ to an intermediate phase to Pd₉Te₄ to PdTe to PdTe₂. However, I see neither science nor engineering. In the following, we provide more detailed comments.”*

Response: We greatly appreciate Reviewer #2’s summary of our work. An introduction regarding the scientific and engineering aspects of this work has been added to the revised manuscript. Precisely controlling the phases in low-dimensional materials enables the tailoring of material properties, resulting in enhanced performance and multifunctionality. Consequently, promoting phase-engineering techniques are of utmost importance. It is imperative to develop novel mechanisms and experimental tools for uncovering and governing phase changes at the nanoscale. However, owing to the intricate thermodynamic and kinetic mechanisms, phase engineering in nanomaterials generally involves only a restricted number of phases and is confined to nano- or micrometer scales, which substantially impedes the diversity and scalability of applications.

In our work, we showcase the exploration of unlocking the stoichiometry as a new degree of freedom for phase engineering in the Pd-Te binary compound. Our discoveries provide valuable perspectives into the mechanism of phase transition via stoichiometry engineering, which exemplifies the scientific nature of this work. By designing the growth structure to achieve spatially uniform nucleation and halt the phase transition at precise points, we attain stoichiometry-controllable wafer-scale growth, opening up novel pathways for expanding the phase library in nanomaterials and facilitating scalable applications, thus demonstrating the engineering aspect. Relevant research has also attracted extensive attention within the field. For instance, Xiaowei Liu and colleagues published an approach that permits in situ tuning between 2D phases of different stoichiometries very recently (Nat. Mater. 23, 1363–1369 (2024)) during our submissions.

Below, we also offer a point-by-point response to the remaining points raised by the reviewer, with the corresponding alterations highlighted in red in the revised manuscript

and/or Supplementary Information.

Comment: *“With phases reported in the manuscript, I do not see evidence of “alloy” as PdyTex the authors reported are actually compounds with crystal structures that do not resemble to either Pd or Te. In addition, all compounds (PdyTex) were not engineered but rather a diffusion process. Therefore, I believe that the title is misleading.”*

Response: We thank the reviewer for raising this suggestion. The term "Alloy" herein refers to the compounds composed of Pd and Te elements. Following the reviewer's suggestion and to avoid ambiguity, we have replaced "alloy" with "compound" in revised manuscript.

As the reviewer pointed out, several phases were obtained via a diffusion process, based on which we can explore the growth kinetics and mechanism. Then, through the designed growth approach, we managed to controllably fabricate wafer-scale phases, highlighting the "engineering" feature of this research. Through engineering (1) the spatially homogeneous nucleation and (2) the final formed phase, we achieved stoichiometry-controllable wafer-scale growth. A homogeneous nucleation process can be achieved by utilizing another substrate coated with a Te film and placed it face-to-face with the Pd substrate to uniformly supply Te (upper panel in Fig. R1).. By precisely adjusting the thickness of the Te film, we were able to control the amount of Te involved in the reaction. Since the stoichiometry transition proceeds in the direction of increasing Te content, we could thus control the cessation of the phase transition and, as a result, achieve stoichiometry-controllable growth.

Fig. R1 Schematic diagram illustrating the stoichiometry-controllable synthesis of wafer-scale Pd-Te films.

Comment: “Identifying five Pd_yTe_x phases does not mean that they are purposely engineered. Instead, it implies that different Pd_yTe_x phase requires different reaction conditions.”

Response: We thank the reviewer for raising this comment. As the reviewer mentioned, we have identified five intermixed Pd_yTe_x phases during the diffusion process (Fig. 2f-i). To further achieve wafer-scale uniform and phase-controllable growth, we need to accomplish: (1) spatially homogeneous nucleation and (2) the capacity to control the final formed phase. Therefore, we employed another substrate coated with a Te film and place it face-to-face with the Pd substrate to uniformly supply Te (upper panel in Fig. R1). Then, a homogeneous nucleation process is observed through the use of an in-situ CVD system. By accurately adjusting the thickness of the Te film, we are able to control the amount of Te participating in the reaction. Since the stoichiometry transition occurs in the direction of increasing Te content, we can thus control the termination of the phase transition and, consequently, achieve stoichiometry-controllable growth. Figure R2 shows the phase diagram of the Pd-Te compound. The detailed description of the overall process is also added in the revised manuscript in page 2.

Fig. R2 The phase diagram of the Pd-Te compound. It shows that different stoichiometries can be obtained under different thicknesses of Pd and Te.

Comment: “References are incorrectly cited. For example, references 41-43 have nothing to do with PdTe_x. On the other hand, literature related to PdTe and PdTe₂ are not cited.”

Response: We thank the reviewer for pointing this out. Following the reviewer's suggestion, we have carefully examined the citations throughout the text and added references related to the work on PdTe and PdTe₂. Regarding references 41-43, although the articles mainly studied the Josephson effect observed in WTe₂, the authors hold the view that the superconducting characteristics originate from PdTe_x which is formed by the reaction between Pd in the electrodes and Te in WTe₂ during the annealing process. However, the specific phases formed and the direct correlation between their superconducting properties and specific phases remain unclear. This ambiguity impedes the investigation of the physical properties and the application of devices based on the Pd-Te films. In our work, by combining controllable phase preparation and low-temperature transport measurements, we directly addressed this issue. Some references related to PdTe and PdTe₂ have also been added in the revised manuscript as Ref. 28-31 and 48-51.

Responses to Reviewer #3

General comments: *“The manuscript titled “Stoichiometry-engineered phase transition in a two-dimensional binary alloy” introduces stoichiometry as a novel approach to phase engineering in Pd-Te binary alloy, enabling controlled wafer-scale growth through reduced diffusion rates and precise nucleation. This study identifies five distinct phases, including four with superconducting properties, offering valuable insights into phase transition and advancing scalable applications in nanomaterials.*

I find this manuscript to be a well-presented study. The authors have clearly articulated their findings in a well-structured organization. The manuscript provides sufficient background information, along with a detailed and comprehensive description of the methodology used. This attention to clarity and detail enhances the readability and reproducibility of the research.”

Response: We greatly appreciate Reviewer #3's precise summary of our work and the positive evaluation. We are encouraged that the Reviewer found that *“offering valuable insights into phase transition and advancing scalable applications in nanomaterials”* and *“this manuscript to be a well-presented study. The authors have clearly articulated their findings in a well-structured organization.”*. His/her comments/suggestions provided are very helpful in improving the quality of our manuscript. Below, we provide a point-by-point response to his/her remaining points, with the corresponding changes highlighted in the revised manuscript and/or Supplementary Information.

Comment: *“The manuscript emphasizes the role of the interface in driving phase transitions and influencing nucleation shapes. However, the discussion lacks depth regarding the impact of vertical phase heterogeneity. For example, in Figure 3d, the irregular shape of the intermediate phase is attributed to the polycrystallinity of the Pd₁₀Te₃ background, yet this explanation is not sufficiently detailed. The authors should provide a more thorough analysis of this phenomenon and propose strategies to mitigate vertical heterogeneity. Addressing this issue would be critical for enhancing phase uniformity and the overall quality of the material.”*

Response: We thank the reviewer for raising this profound comment. Since different phases possess distinct lattice structures, the superposition of two different periodic

structures will generate moiré fringes. Therefore, the presence or absence of Moiré patterns in a transmission electron microscope (TEM) can be used to confirm the vertical phase heterogeneity. From the results of the aberration-corrected STEM as shown in Fig. R1, each phase exhibits a clear atomic structure that is consistent with the lattice structure of that phase, and there are no obvious Moiré fringes, indicating that each phase we grew is uniform in the vertical direction.

Fig. R1 Atomic-resolution STEM images of different nucleation sites and the corresponding simulation results confirm that the nucleation sites correspond to (a-b) $\text{Pd}_{10}\text{Te}_3$, (c-d) Pd_9Te_4 , (e-f) PdTe , and (g-h) PdTe_2 .

To further confirm this, we examined the cross-sectional STEM of the PdTe and Pd_9Te_4 interfaces (Fig. R2). The two phases are uniform in the vertical direction, and the interface between the two phases is very sharp, suggesting that this phase transition is complete in the vertical direction and the phase transition rate is also consistent in the vertical direction.

Fig. R2 Cross-sectional STEM characterization of the interface between the PdTe and Pd₉Te₄. (a) The cross-sectional STEM image at low magnification. (b) The atomic-resolved cross-sectional STEM image. It shows the phase transition is complete in the vertical direction.

Interestingly, after carefully examining the STEM results, we found a Moiré fringe (marked by white dotted lines) at the interface between Pd₁₀Te₃ and intermediate phase (Fig. R3), which should be due to the superposition of the two phases in the vertical direction. However, the region of the moiré pattern is very narrow, only 2 nm in width.

Therefore, from the above in-plane and cross-sectional STEM characterization results, it can be demonstrated that the films we grew have very good uniformity in the vertical direction.

Fig. R3 The moiré fringes (marked by white dotted lines) at the interface between Pd₁₀Te₃ and intermediate phase.

Then, according to the kinetic Wulff construction, the shape of the nucleation is

determined by interface energy and the relative growth rate of different facets. As $\text{Pd}_{10}\text{Te}_3$ is a polycrystal with a grain size of several tens of nanometers, the intermediate phase will encounter grains with various orientations during the phase transition process, thus forming different heterogeneous interfaces. These heterogeneous interfaces will have different growth rates, leading to the irregular shape of the intermediate phase.

These corresponding discussions have been incorporated into the main text in page 7, and Fig R2 and Fig R3 were also added as Extended Data Fig. 7a-b, and Extended Data Fig. 3b in revised manuscript.

Comment: “Regarding the vertical phase heterogeneity mentioned earlier, the authors claim to achieve pure single phases by controlling the Pd and Te thickness. However, while optical microscopy (OM) observations (Figure 4) and video evidence are provided, these alone may not be sufficient to confirm a single phase across the entire wafer. I suggest the authors include XRD measurements to verify this claim.”

Response: We appreciated the reviewer for raising this constructive suggestion. Following the reviewer’s suggestion, to confirm a single phase across the entire wafer, we carried out characterizations of XRD, EBSD, STEM and Raman characterizations for the wafers of four phases.

The homogeneity within a large range of each phase was verified by an X-ray diffractometer (XRD) using $\text{Cu K}\alpha$ radiation. The (0 0 l) peaks of PdTe_2 and PdTe phases indicate that the natural surface is the *a-b* plane (Fig. R4), which is consistent with the EBSD and STEM results.

Fig. R4 The XRD spectra of (a) PdTe₂ and (b) PdTe.

However, for the Pd₉Te₄ and Pd₁₀Te₃ phases, no obvious diffraction peaks were observed, probably due to the thin film thickness and the small grain sizes. Therefore, we firstly characterized different positions of the Pd₉Te₄ wafer using EBSD (Fig. R5). Clear Kikuchi pattern diffractions could be seen at different positions, and they could be well fitted with the lattice of Pd₉Te₄, indicating that the films at different positions had good crystallinity and uniformity.

Fig. R5 Kikuchi pattern diffractions of the Pd₉Te₄ wafer using EBSD. (a) Optical images of Pd₉Te₄ wafer. (b) Clear Kikuchi pattern diffractions could be seen at different positions, indicating that the films at different positions had good crystallinity.

Since the grain size of Pd₁₀Te₃ was too small, we performed STEM characterization on the Pd₁₀Te₃ wafer (Fig. R6). A distinct polycrystalline structure could be seen, with the grain size being in the range of tens of nanometers. When a single grain was magnified, a clear atomic structure that matched well with Pd₁₀Te₃ could be observed.

Fig. R6 STEM characterization of $\text{Pd}_{10}\text{Te}_3$ film. (a) Low-magnification STEM image of polycrystalline $\text{Pd}_{10}\text{Te}_3$. (b) Atomic-resolution HAADF-STEM images of $\text{Pd}_{10}\text{Te}_3$ at different position.

We also characterized the Raman spectra of the grown PdTe and PdTe₂ wafers (Fig. R7), and the Raman signals at different positions were uniform. All the above characterization results indicate that the large-area Pd-Te films we grew possess favorable uniformity.

The characterizations related to the homogeneity of wafers of each phase have been added to the revised Supplementary Information (Fig. S5-9). These corresponding discussions have been incorporated into the revised manuscript (page 10).

Fig. R7 The Raman spectra of PdTe and PdTe₂ wafer. (a-b) optical micrograph of PdTe wafer and the corresponding Raman spectra. (c-d) optical micrograph of PdTe₂ wafer and the corresponding Raman spectra. The Raman characteristic peaks indicate the film has good uniformity over a large scale.

Comment: “*The manuscript lacks sufficient detail on phase stability, which is crucial for practical applications. The authors should elaborate on the thermodynamic and operational stability of each phase to enhance the study's relevance and applicability.*”

Response: We thank the reviewer for pointing this out. To confirm the thermodynamic and operational stability of each phase, we annealed the grown Pd-Te thin films with different phases at 500°C in the absence of Te (Fig. R8). It was found that there was no significant contrast change in the films before and after annealing. The PdTe and PdTe₂ phases were characterized by Raman spectroscopy, and it showed that there was no obvious change in the peak positions and intensities of the films. These results indicate that the films we grew possess high thermal stability.

Fig. R8 The thermodynamic stability of each phase. (a-d) optical micrograph of the sample before annealing and the corresponding Raman spectra. (e-h) optical micrograph of the sample after annealing at 500°C and the corresponding Raman spectra.

In addition, we re-performed the superconductivity test on the device that had been stored in air for one year (Fig. R9). The device still maintained its superconducting characteristics, and there was almost no significant change in the superconducting transition temperature. This further demonstrates the good stability of the grown Pd-Te thin films.

Fig. R9 Variable temperature resistances of as-grown and 1-year-aged PdTe.

These corresponding discussions have also been incorporated into the revised manuscript (page 5 and 12). And the Figs. R8-9 about thermal stability have been added into Supplementary Information (Fig. S2 and 11).

Comment: “The manuscript mentions that the T_c of the four superconducting phases does not exhibit a monotonic trend, but it does not provide sufficient explanation for this behavior. Additionally, while PdTe is highlighted as having the highest density of states (DOS), contributing to its higher T_c , the DOS values for the other phases are not discussed. Including these values and analyzing their relationship with T_c could provide valuable insights into the observed trend and improve the overall depth of the study”

Response: We thank this reviewer for proposing this valuable suggestion. To provide a theoretical understanding to the T_c of the four superconducting phases. We performed the first-principle calculations to obtain the DOS values of the four phases. According to the Bardeen-Cooper-Schrieffer (BCS) theory, the superconducting transition temperature is closely related to the electronic density of states at the Fermi level. A higher density of states means that there are more electronic states near the Fermi level available for electrons to form Cooper pairs. When electrons form Cooper pairs through the interaction of exchanging phonons, a higher density of states provides more possibilities for the formation of electron pairs, which is conducive to the occurrence of superconductivity. In our case, the PdTe has the largest density of states at the Fermi level, thus having the highest superconducting transition temperature. The density of states of other phases is in the order of PdTe₂, Pd₁₀Te₃, and Pd₉Te₄ (Fig. R10), which is consistent with the superconducting transition temperature.

These corresponding discussions have been incorporated into the revised manuscript (page 12). And the Fig. R10 has been added into Supplementary Information (Fig. S12).

Fig. R10 The density of states of Pd₁₀Te₃, Pd₉Te₄, PdTe and PdTe₂.

Responses to Reviewer #1

General comments: *“The author had adequately addressed all of my questions and comments. It is particularly remarkable that a similar mechanism observed in the Mo-Te system. I believe this manuscript is suitable for publication in Nature Communications without any further revisions.”*

Response: We are extremely grateful for your positive evaluation and kind words. We are delighted that the manuscript has met your high standards. Thank you again for your support and valuable time dedicated to reviewing our work.

Responses to Reviewer #2

General comments: “The revised manuscript has clarified some aspects of concerns. However, the overall story is about the formation of five phases. It still lacks basic information.”

Response: We sincerely appreciate the Reviewer#2’s time and effort in reviewing our revised manuscript. The feedback is of great value to us, and we are glad that some of the concerns have been clarified. Below, we provide a point-by-point response to his/her remaining points, with the corresponding changes highlighted in the main text and/or Supplementary Material in the revised version of our manuscript.

Comments: “For example, while Figure 1 shows the phase distribution through the process, these phases are later characterized via EDX, STEM, SAED, EBSD, and simulation. However, there is no information about the crystal structure and corresponding lattice parameters of each phase. The lack of such information makes readers difficult to follow the description. In line#203, the authors state that “Due to the low symmetry of Pd₉Te₄ lattice, two different interfaces are formed with PdTe …”. Without knowing the symmetry of Pd₉Te₄ and PdTe, one cannot follow the discussion. The authors should be aware that similar losing description occurs in multiple places.”

Response: We thank the reviewer for the meticulous review and constructive feedback on our manuscript. We are committed to addressing them comprehensively to enhance the clarity and comprehensibility of our work.

Regarding the missing details on crystal structures and lattice parameters of each phase, we have conducted in-depth research and compiled detailed tables. These tables will be added to the revised supplementary information file, providing the crystal structure type (such as monoclinic, hexagonal, etc.), space group and the corresponding lattice parameters (a, b, c, α , β , γ) for each of the phases involved. This addition will enable readers to better visualize and understand the phases described in the manuscript.

[REDACTED]

[REDACTED]

[REDACTED]

[REDACTED]

[REDACTED]

[REDACTED]

[REDACTED]

[REDACTED]

[REDACTED]

[REDACTED]

[REDACTED]

[REDACTED]

[REDACTED]

In response to the statement about the interfaces formed by Pd₉Te₄ and PdTe in line #203, we will add information in the manuscript that elaborates on the symmetry of Pd₉Te₄ and PdTe. We then explain how the low symmetry of Pd₉Te₄ leads to the formation of two different interfaces with PdTe. This will make the discussion more accessible and follow-able for the readers.

We will conduct a thorough review of the entire manuscript, identify all such areas, and add the necessary information and explanations to ensure a coherent and easy-to-follow narrative.

Comment: *“In view of Fig. 1(f), this reviewer cannot tell the difference between spectra for three phases.”*

Response: Thank you for pointing out this issue. Regarding the comment about not being able to tell the difference between the spectra for the three phases in Fig. 1(f), we would like to explain that in our experimental process, no obvious Raman signals were collected for any of these three phases.

The absence of distinct Raman signals could be attributed to several factors. First, experimental conditions such as the laser power, spectra range, resolution and the interaction time between the laser and the sample may not be optimized for detecting strong Raman signals from these particular phases. Second, the chemical bonds and molecular structures of these phases might result in relatively weak Raman scattering cross-sections.

To address this concern and to help readers better understand the characteristics of these phases, we will add a detailed note in the figure caption of Fig. 1(f). The note will clearly state that no obvious Raman signals were obtained for the three phases

and briefly mention the possible reasons as explained above.

We optimized the testing conditions (laser power, focusing, exposure time, etc.) and conducted Raman spectra measurement on the Pd₁₀Te₃ phase, intermediate phase, and Pd₉Te₄ phase within the range from 50 cm⁻¹ to 2000 cm⁻¹ as shown in Fig. R1. However, apart from the silicon Raman peak of the substrate, no distinct Raman peaks were detected. To avoid the omission of Raman peaks due to resolution, we also carried out measurements using a grating with 2400 g/mm. Nevertheless, no significant Raman peak was obtained. Since both Pd and Te elements are relatively heavy, we believe that the Raman signals of these phases will not exceed 2000 cm⁻¹. The cross section of Raman scattering is so low that usually only one Raman photon is produced for every 10⁶-10¹⁰ incident photons for different materials (Overview of Raman Spectroscopy: Fundamental to Applications. *Modern Techniques of Spectroscopy*, Springer, 13, 145–184, 2021). The cross section of Raman scattering of these three phases might be too small, leading to a relatively weak Raman signals. Due to the absence of distinct Raman peaks, we did not find any differences in the Raman spectra curves of the three phases. The identification of different phases was achieved through subsequent STEM characterization and simulations.

Fig. R1 The Raman spectra of $\text{Pd}_{10}\text{Te}_3$ at (a) 600 g/mm and (b) 2400 g/mm. The Raman spectra of intermediate phases at (c) 600 g/mm and (d) 2400 g/mm. The Raman spectra of Pd_9Te_4 at (e) 600 g/mm and (f) 2400 g/mm. No obvious Raman peaks are observed.

Comment: “Regarding the observation of superconductivity, the authors should be aware that superconductivity has been heavily studied in bulk PdTe and PdTe_2 . The relevant references have not been included even in the revised manuscript.”

Response: We sincerely appreciate your valuable feedback regarding the references on superconductivity in bulk PdTe and PdTe_2 . Following the reviewer's suggestion, we have identified several key studies on superconductivity in bulk PdTe and PdTe_2 that are highly relevant to our research. These references will be carefully integrated into the manuscript as Ref. 52-67 in the discussion section.

Reference

1. V.S. Khar'kin, R.M. Imamov, and S.A. Semiletov. The palladium telluride Pd_{4-x}Te

crystal structure. *Kristallografiya*, 14 :907–909, 1969.

2. W.S. Kim, G.Y. Chao, and L.J. Cabri. Phase relations in the Pd-Te system. *Journal of the Less-Common Metals*, 162:61–74, 1990.
3. P. Matkovic and K. Schubert. Kristallstruktur von Pd₉Te₄. *Journal of the Less-Common Metals*, 58:39–46, 1978.
4. L.J. Cabri, J.F. Rowland, J.H.G. Laflamme, and J.M. Stewart. Keithconnite, telluropalladinite and other palladium-platinum tellurides from the stillwater complex, montana. *Canadian Mineralogist*, 17:589–594, 1979.
5. A. Kjekshus and W.B. Pearson. Constitution and magnetic and electrical properties of palladium tellurides (PdTe - PdTe₂). *Canadian Journal of Physics*, 43:438–449, 1965.
6. L. Thomassen. Ueber kristallstrukturen einiger binaerer verbindungen der platinmetalle. *Zeitschrift fuer Physikalische Chemie, Abteilung B: Chemie der Elementarprozesse, Aufbau der Materie*, 2:349–379, 1929.
7. F. Gronvold and E. Rost. On the sulfides, selenides and tellurides of palladium. *Acta Chemica Scandinavica (1-27,1973-42,1988)*, 10:1620–1634, 1956.
8. W.O.J. Groeneveld Meijer. Synthesis, structures, and properties of platinum metal tellurides. *American Mineralogist*, 40:646–657, 1955.
- 9 S. Furuseth, K. Selte, and A. Kjekshus. Redetermined crystal structures of NiTe₂, PdTe₂, PtS₂, PtSe₂, and PtTe₂. *Acta Chemica Scandinavica (1-27,1973-42,1988)*, 19:257–258, 1965.
10. E. McCarron, R. Korenstein, and A. Wold. High pressure phase transformation studies of the system Pd_{1-x}Rh_xTe₂. *Materials Research Bulletin*, 11:1457–1462, 1976.
11. A. Kjekshus and F. Gronvold. High temperature X-ray study of the thermal expansion of PtS₂, PtSe₂, and PdTe₂. *Acta Chemica Scandinavica (1-27,1973-42,1988)*, 13:1767–1774, 1959.
12. M.A. Pell, Yu.V. Mironov, and J.A. Ibers. PdTe₂. *Acta Crystallographica, Section C: Crystal Structure Communications*, 52:1331–1332, 1996.

Responses to Reviewer #3

General comments: “I appreciate the authors' efforts to address the comments from the first round of review. The revisions have significantly improved the manuscript, and most of my concerns have been satisfactorily addressed.

In summary, the authors have made substantial improvements to the manuscript, and the additional data provided are sufficient to address most of my initial concerns. I recommend minor revisions to address the remaining points regarding strain and phase stability at higher temperatures.”

Response: We greatly appreciate Reviewer #3's positive evaluation. We are encouraged that the reviewer found that “*The revisions have significantly improved the manuscript, and most of my concerns have been satisfactorily addressed.*”. His/her comments/suggestions are very helpful in improving the quality of our manuscript. Below, we provide a point-by-point response to his/her remaining points, with the corresponding changes highlighted in the revised manuscript and/or Supplementary Information.

Comment: “*Confirmation of Single Phase Across Wafer: The authors followed my suggestion by including XRD and Raman data for PdTe and PdTe₂, as well as EBSD for Pd₉Te₄ and STEM for Pd₁₀Te₃. This comprehensive data set confirms the presence of distinct phases across the wafer. However, I am curious whether the coexistence of multiple phases on the same wafer induces strain. Could the authors comment on whether strain arises in the mixed-phase samples due to the expansion during tellurization?*”

Response: We extend our gratitude to the reviewer for raising this detailed concern. In order to confirm whether there is strain in the mixed-phase samples, we conducted a strain analysis on the STEM images. Initially, through Geometrical Phase Analysis (GPA), we calculated the strain distribution for each individual phase, as illustrated in Figures R1-4, which depicts the strain in the X-direction, Y-direction, and shear strain for Pd₁₀Te₃, Pd₉Te₄, PdTe, and PdTe₂, respectively. Although the strain distribution in different regions shows certain differences, and there are cases of tension or compression, these strains are basically around 0.

Figure R1 STEM image and GPA analysis of Pd₁₀Te₃.

Figure R2 STEM image and GPA analysis of Pd₉Te₄.

Figure R3 STEM image and GPA analysis of PdTe.

Figure R4 STEM image and GPA analysis of PdTe₂.

Taking Pd₁₀Te₃, which has a relatively simple top-view atomic image, as an example, we conducted an analysis of the atomic phase strain. The primary methodology is as follows: Python scripts were used to calculate atomic strain. The atomic positions in HAADF images were located using the peak detection algorithm. The centrefold fitting and Gaussian fitting were employed to further improve the localization

accuracy. By constraining the angle and distance, the coordinates of adjacent atom in specified direction for each atom was determined. The distance between an atom and its adjacent atoms was calculated using the formula $d = \sqrt{(x - x_0)^2 + (y - y_0)^2}$, where d represents the distance, and (x, y) , (x_0, y_0) denote the coordinates of the atom and its adjacent atoms respectively. The obtained data were compared with the standard atomic structure to determine the value of atomic strain, which was visualized by the Matplotlib library.

The results are depicted in Figures R5, which present the atomic strain analysis for $\text{Pd}_{10}\text{Te}_3$. We statistically analyzed the strain distribution in direction1 (labeled by arrow). For $\text{Pd}_{10}\text{Te}_3$, the strain was minimal, manifesting as tensile strain near 0%. This result is consistent with that of the GPA.

Figure R5 The atomic strain in $\text{Pd}_{10}\text{Te}_3$

Given the top-view atomic images underwent a transfer process from the substrate to the copper micro-grid, potential strain release may have influenced the final strain calculation outcomes. Consequently, we also performed calculations on the side-view atomic image without transfer, as shown in Figure R6, which presents the strain distribution at the interface between Pd_9Te_4 and PdTe . The findings reveal that strain is predominantly present at the interfaces between phases, with minimal strain observed within individual phases.

Figure R6 STEM image and GPA analysis of side-view interface between Pd₉Te₄ and PdTe.

We hypothesize that the strain generated during the tellurization is mainly released through expansion in the vertical direction. In addition, some local strain is released by the formation of small crystals, which is consistent with our atomic force microscopy (AFM) results that the thickness has increased compared with that of the Pd film and roughness gradually increase as the thickness of the Pd film increases.

Comment: “*Phase Stability: The authors have demonstrated good thermodynamic stability by annealing the samples at 500°C without Te, with no significant changes observed in OM and Raman data. Additionally, the superconductivity test on a device stored for one year confirms long-term stability. However, I would like to know at what temperature the stability is lost. Since Te typically degrades around 550°C, will the compound undergo phase transitions above this temperature?*”

Response: Thank the reviewers for raising the detailed questions. To confirm what temperature the stability would be lost, we annealed the grown Pd-Te thin films with different phases at 500 and 550°C in the absence of Te (Figure R7). It was found that there was no significant contrast change in the films after annealing at 500°C. We further increased the temperature to 550°C for approximately 30 minutes, as the reviewers predicted, the film decomposed and lost its Raman signal. So, the film loses its stability between 500-550°C. Te or Pd is essential to stabilize and trigger the phase transition.

Figure R7 The thermodynamic stability. (a, b) optical micrograph of the sample before annealing and the corresponding Raman spectra. (c, d) optical micrograph of the sample after annealing at 500°C and the corresponding Raman spectra. (e, f) optical micrograph of the sample after annealing at 550°C and the corresponding Raman spectra.